# Integrated polarization-sensitive amplification system for digital information transmission

Wenhao Ran[1,2], Zhihui Ren[1,2], Pan Wang[1,2], Yongxu Yan[1,2], Kai Zhao[1,2], Linlin Li[1,2], Zhexin Li[1,2], Lili Wang[1,2], Juehan Yang[1,2], Zhongming Wei [1,2✉], Zheng Lou [1,2✉] & Guozhen Shen [1,2✉]

Polarized light can provide significant information about objects, and can be used as information carrier in communication systems through artificial modulation. However, traditional polarized light detection systems integrate polarizers and various functional circuits in addition to detectors, and are supplemented by complex encoding and decoding algorithms. Although the in-plane anisotropy of low-dimensional materials can be utilized to manufacture polarization-sensitive photodetectors without polarizers, the low anisotropic photocurrent ratio makes it impossible to realize digital output of polarized information. In this study, we propose an integrated polarization-sensitive amplification system by introducing a nanowire polarized photodetector and organic semiconductor transistors, which can boost the polarization sensitivity from 1.24 to 375. Especially, integrated systems are universal in that the systems can increase the anisotropic photocurrent ratio of any low-dimensional material corresponding to the polarized light. Consequently, a simple digital polarized light communication system can be realized based on this integrated system, which achieves certain information disguising and confidentiality effects.

[1] State Key Laboratory of Superlattices and Microstructures, Institute of Semiconductors, Chinese Academy of Sciences, Beijing 100083, China. [2] Center of Materials Science and Optoelectronic Engineering, University of Chinese Academy of Sciences, Beijing 100049, China. ✉email: zmwei@semi.ac.cn; zlou@semi.ac.cn; gzshen@semi.ac.cn

The polarization state of light carries a large amount of effective information, which cannot be obtained through the intensity, wavelength, frequency, and phase. This information can be utilized to realize the information transmission through artificial modulation[1,2]. The use of polarization-sensitive photodetectors to achieve effective detection of polarized light can enhance the information of detected objects, significantly improving the target contrast. This promotes detection and imaging, and polarized light communication can thus be realized to achieve confidential information transmission[3,4]. Therefore, polarization-sensitive photodetectors are widely utilized in various fields and are of great importance for industrial applications, national security, biological diagnosis, and astronomy[5,6]. Such applications require high polarization sensitivity to determine the number of effectively-recognized polarization angles. However, the structure of traditional detection systems for polarized light detection is complicated and requires prepositive polarizer arrays supplemented by filters, amplifying circuits, and complex decoding algorithms for outputting digital information. This makes such systems bulky and limits the scope for miniaturized integration in applications.

In recent years, the rise of low-dimensional (LD) materials has opened up another way for high-performance photodetection, which benefits from the limit size limitation in thickness and the strong light–matter interaction[7–11]. More importantly, some LD materials with intrinsic in-plane anisotropic structures can potentially detect polarized light without polarizers[5,12,13]. Therefore, it is feasible to develop advanced polarization detection devices using in-plane anisotropic semiconductors. Although many LD materials have been successfully prepared for polarization-type photodetectors, there are some limiting factors such as instability under environmental conditions, narrow optical response range, and insufficient anisotropic photocurrent rate (<10). These lead to the inability to achieve digital output currently. Most existing methods focus on improving the anisotropic photocurrent rate from the perspective of designing anisotropic materials, such as constructing a variety of van der Waals heterojunctions and integrating nano-antennas on the surfaces of LD materials[12]. However, these methods are not universal and introduce challenges in preparation techniques, device integration, and processing compatibility. Therefore, a universal method to improve polarization sensitivity is highly desirable for the accurate identification and digitization of multiple information in polarized optical communication.

In this study, we propose a simple and universal polarization-sensitive amplification system (PSAS) to improve the anisotropic photocurrent rate of LD materials. The proposed system realizes high-performance polarized light coding recognition and digital output. A $C_8$-BTBT/PS-based organic field-effect transistor (OFET) was integrated into a polarization-sensitive $Bi_2Se_2S$ nanowire (NW) photodetector, constructed a polarization-sensitive amplification system, which enhances the polarization sensitivity of the system by several orders of magnitude. The proposed system exhibited an effective increase in the anisotropic photocurrent ratio of the NW photodetector from 1.24 to 375 under 532 nm light irradiation. Furthermore, the anisotropic photocurrent ratios of the other wavelengths show the same significant improvement. To reflect the advantages of the increased anisotropic photocurrent ratio, we compared the accuracy of image recognition between the proposed model and that using an artificial neural network (ANN). As a proof-of-concept, we connect the system to the designed circuit to demonstrate the transmission of information through polarized light and use our integrated system for detection to realize the digital output of polarized optical information and information disguising.

## Results

**Device architecture.** Digitally encoding information can effectively increase the speed and quantity of information in transmission. The use of polarized light to encode information exhibits these advantages and introduces certain information confidentiality and camouflage effects. Polarization-sensitive photodetectors are the core components of polarized optical communication systems, and their performances directly determine the quality of information transmission of the system. Polarization-sensitive photodetectors based on LD materials are used in such systems, owing to their unique response to polarized light. However, the low anisotropic photocurrent ratio results in the use of filtering and amplifying circuits to process the output signal. This process is prone to errors in the digitization process, which seriously reduces the quality of information transmission. To address these issues, we designed a PSAS. Figure 1a shows a schematic diagram of the information transmission results based on the PS and PSAS. A series of linearly polarized light beams with different polarization angles of 0°, 30°, 60°, and 90° were utilized to encode the sending pattern. The PS and PSAS detect signals separately to perform analog-to-digital conversions. The PS-based system does not reproduce the information correctly due to the low anisotropic photocurrent ratio. In contrast, the information can be effectively reproduced by the PSAS, because the anisotropic photocurrent ratio is significantly improved. Thus, the correct signal output can be achieved.

Figure 1b shows the detailed structure of the PSAS, which is comprised of three main parts: the detector based on the $Bi_2Se_2S$ NW used to detect polarized light, the reference resistor used to divide the voltage, and the transistor based on the $C_8$-BTBT/PS used to amplify the signal. The gate in the transistor unit is replaced by an electrode of the photodetector and the reference resistor, constituting an integrated amplifier circuit. The working principle of the PSAS is divided into the following three main processes: (i) Only intrinsic carriers are present in the $Bi_2Se_2S$ NW without illumination with low carrier concentrations, resulting in high resistance of the $Bi_2Se_2S$ NW-based PS and a reference resistor. This further leads to a higher voltage being applied to the gate electrode of the transistor, switching the transistor to the off-state. (ii) With polarized light illuminating the photodetector, $Bi_2Se_2S$ NWs generate photo-generated carriers, leading to a decrease in the resistance of the photodetector. This increases the gate electrode voltage of the transistor. The transistor is partially turned on and outputs a higher current at this instant. (iii) The resistance of the photodetector changes with the polarization angle, resulting in a slight variation in the gate electrode voltage. Since the transistor operates near the threshold voltage, a small change in the gate electrode voltage leads to a significant change in the output current, thereby increasing the anisotropic photocurrent ratio. After removing the polarized light, the integrated system is restored to its initial state.

The basic block diagram of the polarized optical communication system based on conventional polarization-sensitive photodetectors, PS and PSAS, are shown in Fig. 1c, d. The three systems are composed of four parts: information sending, information transmission, information receiving, and information processing. In conventional systems (top panel in Fig. 1c), the detector of the information receiving part is comprised of a series of polarizers and a photodetector. The filtering and amplifying circuit is connected to the detector to reduce noise and amplify the detected signal. Then, analog-to-digital conversion and processing is performed to realize digital signal output. While ensuring the quality of information transmission, the system facilitates subsequent information processing. Although the performance of the system is excellent, the structure is significantly complicated.

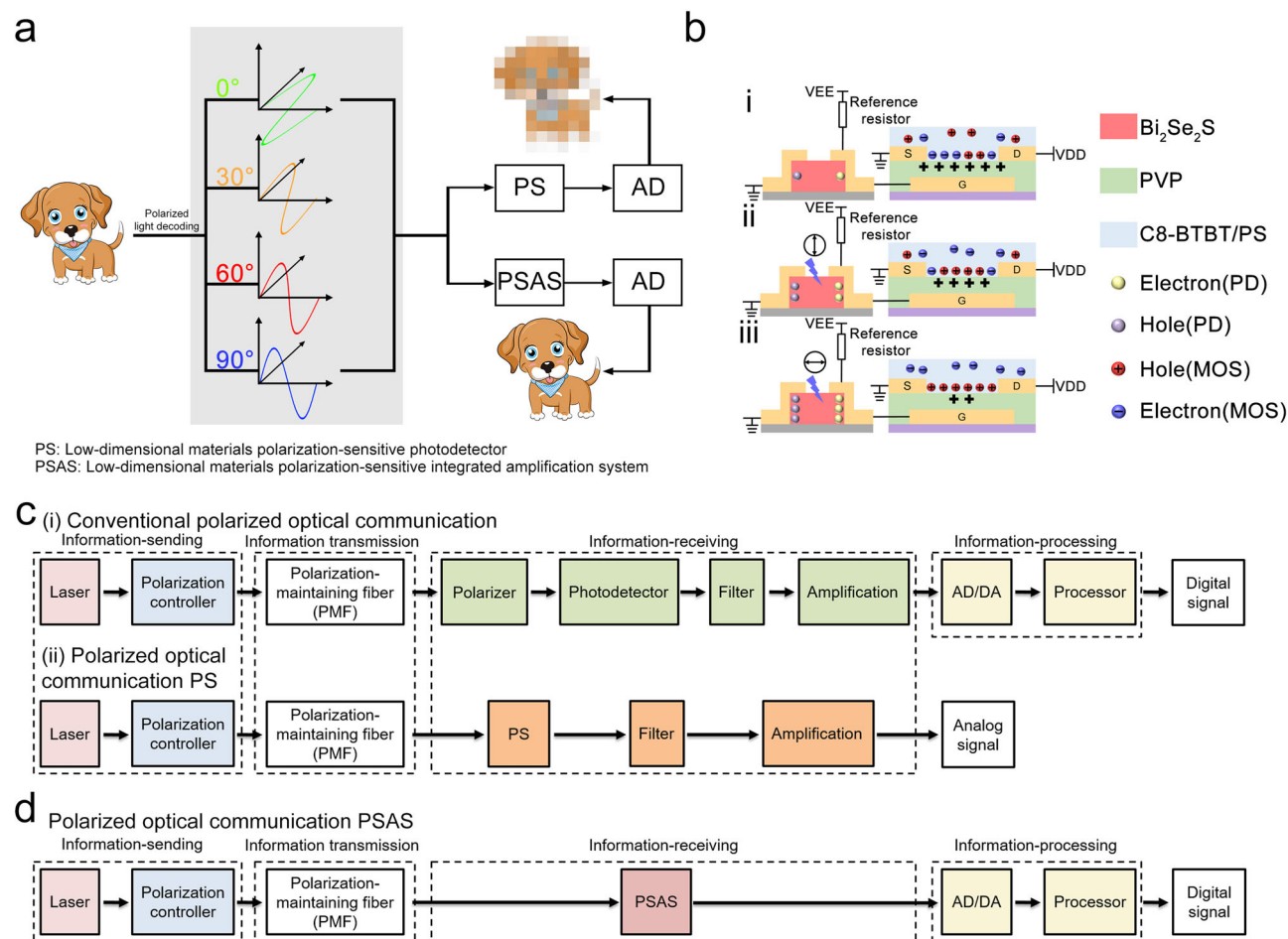

**Fig. 1 Working principle of PSAS and system structure of polarized optical communication. a** Comparison of information transmission results between the PS-based and PSAS-based systems. **b** Working principle and structure schematic diagram of the PSAS. **c** Block diagram showing the sequence of conventional polarized optical communication and PS-based polarized optical communication. **d** Block diagram showing the sequence of PSAS-based polarized optical communication.

Although using PS as a detector can effectively reduce system components, filtering and amplifying circuits are still requirements (bottom panel of Fig. 1c). However, owing to the low anisotropic photocurrent ratio, the digitization process involves significant errors. Therefore, only analog signal output can be realized. In contrast, while ensuring the quality of information transmission, PSAS can effectively simplify the structure and processing of the entire system. Owing to the high anisotropic photocurrent ratio, PSAS does not require a polarizer like in PS and reduces the need for filtering and amplifying circuits. In turn, the system structure can be optimized substantially, and the complexity of the system design can be reduced.

**Material properties and electrical properties of PS-based on Bi₂Se₂S NWs.** $Bi_2Se_2S$ NWs were synthesized by chemical vapor deposition, the crystal quality and elemental composition were verified through various material characterizations (Supplementary Note 1). According to the characterization results, we can find that the $Bi_2Se_2S$ NWs belong to the orthorhombic structure, which is a low-symmetry lattice structure. This low-symmetry crystal structure makes $Bi_2Se_2S$ often grow into an NW morphology, therefore there must be a primary growth direction and a secondary growth direction in the growth plane. This will cause the absorption coefficient of the incident light to be different in the two directions, so there is bound to anisotropy in photoelectric properties. Owing to the unique morphology and the in-

plane anisotropic lattice structure, $Bi_2Se_2S$ NWs with lamellar structures are particularly advantageous as photosensing elements for polarization-sensitive photodetectors. Therefore, we used a single $Bi_2Se_2S$ NW-based PS as the detector of the PSAS. To further analyze the possible optical and electrical properties of the materials theoretically, we constructed a theoretical model and evaluated the $Bi_2Se_2S$ NW. The lattice constants of the relaxed bulk $Bi_2Se_2S$ are a = 11.47 Å, and b = 11.98 Å, c = 4.13 Å, and the band structure was calculated using the HSE06 functional under the integral path of $\Gamma \rightarrow X \rightarrow S \rightarrow Y \rightarrow \Gamma$. As shown in Fig. 2a, the results indicate that $Bi_2Se_2S$ is a direct bandgap semiconductor with a bandgap of 1.18 eV. Therefore, its maximum absorption wavelength is ~1050 nm, which is close to the experimental results. The atom-resolved and total density of states (DOS) of the bulk $Bi_2Se_2S$ are also shown in Supplementary Fig. 4a, which indicates that S and Se atoms mainly contribute to the top of the valence band of $Bi_2Se_2S$, while Bi atoms mainly contribute to the bottom of the conduction band. To investigate the anisotropic absorption of bulk $Bi_2Se_2S$, the partial charge densities of the valence band maximum (VBM) and conduction band minimum (CBM) in the ab-and bc-planes were calculated by VASP, and are shown in Supplementary Fig. 4b. It was found that the charges were mainly localized on the S and Se atoms at the VBM, while they were localized on the Bi atom at the CBM. This is in good agreement with the DOS results. Theoretically, the electric dipole transition probability of photons absorbed per unit time is

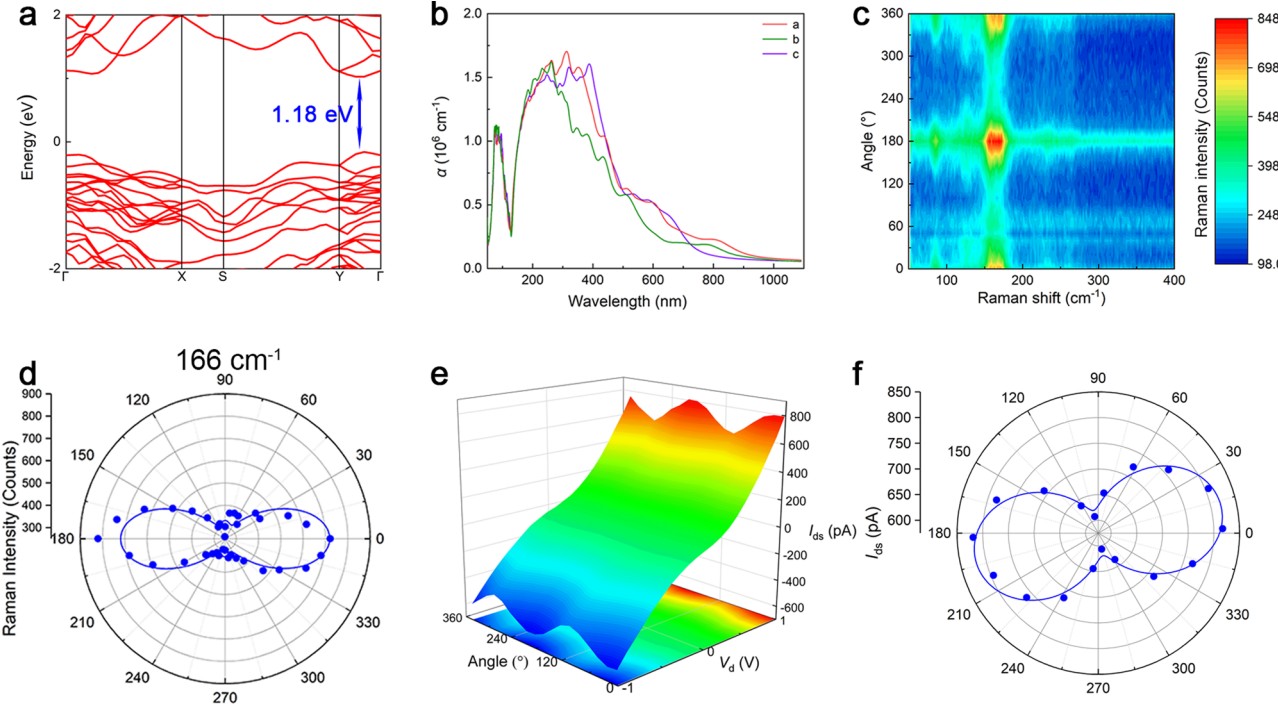

**Fig. 2 Material characterization and the polarized optoelectronic response of Bi$_2$Se$_2$S NW-based PS. a** The band structure of the bulk Bi$_2$Se$_2$S calculated by the HSE06 functional. **b** Calculated optical absorption coefficients α(ω) of Bi$_2$Se$_2$S along the a, b, and c axes. **c** The angle-resolved Raman scattering spectra under parallel configurations of 532 nm laser. **d** Polar plot of Raman peaks at 166 cm$^{-1}$ belonging to the Ag mode under parallel configurations of 532 nm laser. **e** Current-voltage (I-V) curve under 638 nm (optical power density of 111.368 mW/cm$^2$) with different polarization angle. **f** Polar diagram of the relationship between photocurrent and polarization angle under 638 nm at 1.0 V drain bias, the anisotropic photocurrent ratio is 1.32.

proportional to the electron–radiation interaction Hamiltonian matrix element $\langle c|H_{eR}|v \rangle \mid (\langle c|H_{eR}|v \rangle)|$, where $|c\rangle$ and $|v\rangle$ are the wave functions of electrons in the conduction band and valence band, respectively. Therefore, different charge distributions along the a-, b-, and c-axes result in the anisotropic optical properties of Bi$_2$Se$_2$S. The optical absorption spectrum of Bi$_2$Se$_2$S was obtained using first-principle calculations. Based on the crystal planes calibrated in TEM, we can see that Bi$_2$Se$_2$S NWs grow along the a-axis and stack along the b-axis during the growth process. Therefore, when linearly polarized light is irradiated on the NW, the anisotropic photocurrent is mainly due to the difference in the absorption of polarized light by the Bi$_2$Se$_2$S NW in the a-axis and b-axis direction. As shown in Fig. 2b, the absorption coefficient of Bi$_2$Se$_2$S is significantly different along the a and b directions from 300 to 1000 nm. Supplementary Fig. 4c displays the ratio of the absorption coefficient of a-axis and b-axis ($\alpha_a/\alpha_b$), which illustrates a-axis and b-axis have obvious optical anisotropy. In order to further prove that Bi$_2$Se$_2$S NWs have strong anisotropy for light absorption on the a-axis and b-axis, we calculated the probability ($p_a$) of the transition from $|\varphi_{CBM}\rangle$ to $|\varphi_{VBM}\rangle$ along the k-points path, which calculation formula is:

$$p_a = |\langle \varphi_{CBM}|\nabla_a|\varphi_{VBM}\rangle|^2 / \left( |\langle \varphi_{CBM}|\nabla_a|\varphi_{VBM}\rangle|^2 + |\langle \varphi_{CBM}|\nabla_b|\varphi_{VBM}\rangle|^2 \right)$$

(1)

where $\varphi$ represents the wave function, $\nabla$ represents the momentum operator and $|\langle \varphi_{CBM}|\nabla|\varphi_{VBM}\rangle|^2$ is the optical transition amplitude. According to the calculation results in Supplementary Fig. 4d, we can clearly observe that only the absorption of a-polarized light occurs across the gap between Vb$_\Gamma$ and Cb$_\Gamma$ at the $\Gamma$ point ($p_a = 1.0$), while the absorption of b-polarized light is forbidden. Different absorption coefficients indicate that linearly polarized light with different polarization angles exhibits different photoelectric properties when irradiated on the surface of the

material. Thus, Bi$_2$Se$_2$S NWs can be used to fabricate polarization-sensitive photodetectors.

Raman spectroscopy is a powerful technique to determine the vibration modes of various photons in NWs, thereby confirming the optical anisotropy of NWs. Supplementary Fig. 5a shows the traditional unpolarized Raman scattering spectra. Six distinct Raman peaks are observed at 64, 86, 126, 166, 233, and 263 cm$^{-1}$, which are close to the Raman spectra of Bi$_2$Se$_3$[14,15] and Bi$_2$S$_3$[16]. To further explore the anisotropy of photo-vibration and determine the lattice orientation of Bi$_2$Se$_2$S NWs, angle-resolved polarized Raman spectroscopy (ARPRS) was performed under linearly polarized light in parallel and cross configurations at different polarization angles from 0° to 360°. Supplementary Fig. 5b, c exhibit a series of Raman scattering spectra corresponding to different rotation angles in parallel and cross configurations, where the Bi$_2$Se$_2$S NW rotates 10° in-plane each time. The corresponding contour color maps of the two configurations are shown in Fig. 2c and Supplementary Fig. 5d. Evidently, the intensities of the different Raman peaks change regularly with the rotation angle. To demonstrate the anisotropy of the Bi$_2$Se$_2$S NWs more prominently, we extracted the intensities of the Raman peaks under different in-plane rotation angles and constructed the polar coordinate diagram of the Raman peak intensity corresponding to the angle change (Fig. 2d and Supplementary Fig. 6). To verify the growth mode of the Bi$_2$Se$_2$S NWs using the experimental data, we fitted the results to the above polar coordinate diagrams (Supplementary Note 2). By comparing the theoretical curves and experimental data, it can be preliminarily determined that the Raman peaks at 64, 86, and 166 cm$^{-1}$ correspond to the $A_g$ mode, and the peak at 126 cm$^{-1}$ corresponds to the $B_{1g}$ mode. Based on the calculations, we fit the corresponding experimental data in Fig. 2d and Supplementary Fig. 6, which match well with each other. This deviation is mainly due to the interference of the adjacent Raman peaks. In the $A_g$

mode, the curve of the parallel configuration is spindle-shaped, with the strongest Raman scattering intensity at 0°. The second strongest and weakest Raman peaks appear at 90° with a period of 180°. The curve for the cross configuration is four-leaf shaped, with the strongest and weakest Raman peaks at 45° and 0°, respectively. In the $B_{1g}$ mode, the curves are four-leaf-shaped irrespective of the configuration, but the strongest and weakest Raman peaks correspond to opposite polarization angles. In the parallel configuration, the strongest and weakest peaks appear at 45° and 0°, respectively. In the cross configuration, the strongest and weakest peaks appear at 0° and 45°, respectively. In addition to ARPRS, polarization-resolved optical microscopy images are often used to directly observe the anisotropy of the grown materials (Supplementary Fig. 8).

In order to explore the properties of the polarized optoelectronics, we fabricated a polarization-sensitive photodetector based on $Bi_2Se_2S$ NWs. The linear polarization detection measurement system is described in Supplementary Note 3. Since the bandgap of $Bi_2Se_2S$ is 0.8 eV, the highest corresponding wavelength that can be detected is 1550 nm. LD materials are narrower compared with bulk materials, and thus, we selected four wavelengths of 532, 638, and 808 nm for testing. Figure 2e depicts the relationship between the photocurrent and polarization angle of linearly polarized light at different bias voltages ($V_{ds} = -1$ V to 1 V) under 638 nm illumination, with an optical power density of 111.368 mW/cm². From the false-color plot, it can be observed that the photocurrent increases and decreases periodically as the polarization angle changes. The photocurrent reaches its maximum value of ~820 pA when the polarization angle is ~10° or 190°. As $V_{ds} = 1$ V, the photocurrent changes most significantly with the polarization angle. Therefore, we measured the time-resolved photoresponse under different polarization angles at this bias voltage, and the results are shown in Supplementary Fig. 10a. Furthermore, we extracted the polarized photocurrent corresponding to each polarization angle, and expressed using a rectangular coordinate diagram (Supplementary Fig. 10b) and a polar coordinate diagram (Fig. 2f). In order to make the variation rule of the photocurrent in the rectangular coordinate diagram more obvious and the two-circle shape in the polar coordinate diagram more prominent, the formula:

$$I_{ph}(\delta) = I_{px}\sin^2(\delta + \phi) + I_{py}\cos^2(\delta + \phi) \qquad (2)$$

is used to fit the polarized photocurrent, where $\delta$ is the polarization angle, $\Phi$ is the fixed angle between the y-axis and 0°, $I_{ph}(\delta)$ is the photocurrent along the $\delta$ direction, and $I_{px}$ and $I_{py}$ represent the photocurrent along the x- and y-axes, respectively. Apparently, the fitting results match the experimental data well, and the anisotropic photocurrent ratio ($I_{py}/I_{px}$) is about 1.32 at this wavelength. Supplementary Fig. 11 shows the relationship between the polarization photocurrent and the polarization angle of the device at 532 nm (optical power density of 35.873 mW/cm²) and 808 nm (optical power intensity of 53.980 mW/cm²). The corresponding $I_{py}/I_{px}$ values were 1.24 and 1.22, respectively. Supplementary Fig. 12 displays the responding time of $Bi_2Se_2S$ NW-based PS ($\tau_{rise}$ >400 ms, $\tau_{decay}$ <400 ms). Owing to the error in the detected polarization state of the laser with the analyzer, the polarization angle corresponding to the maximum polarization photocurrent exhibits a deviation at a polarization angle of 638 nm. These results indicate that our polarization-sensitive photodetector has better anisotropy, and can effectively distinguish different polarization angles. However, if the photodetector is directly applied to a polarized light communication system and interconnected with an external circuit, the electrical noise in the circuit can easily interfere with the detected signal, causing system disorder.

**Polarized optoelectronic properties of the PSAS.** To make the polarization-sensitive photodetector robust against system noise, improving the anisotropic photocurrent ratio of the device is crucial. Therefore, we designed a PSAS by integrating $Bi_2Se_2S$ NW photodetectors and OFET based on $C_8$-BTBT/PS to improve the anisotropic photocurrent ratio. A schematic of the integrated system is shown in Fig. 3a, including a $Bi_2Se_2S$ NW-based PS, a reference resistor, and a $C_8$-BTBT/PS-based transistor. The physical diagram of the system is illustrated in Supplementary Fig. 13. The system mainly uses the large-signal model of transistors. The transistor with low threshold voltage and low sub-threshold swing are very important for our system. The detailed electric properties of the OFET are described in Supplementary Note 4. $C_8$-BTBT/PS OTET with 1.3 V threshold voltage and 160 mV/decade sub-threshold swing can meet the requirements of PSAS. While realizing the functions, it makes the whole system work at a lower voltage. The response time of less than 25 ms based on the frequency response characteristics makes the C8-BTBT/PS OFET match well with the $Bi_2Se_2S$ NW-based PS. In addition, the solution preparation process of OFET makes the production of the device relatively simple and easy to produce on a large area. The flexible substrate makes our PSAS more flexible in the application process and can adapt to complex environments. Figure 3b shows the polar coordinate diagram of the relationship between the polarized photocurrent and polarization angle of the PSAS at $V_{dd} = -0.5$ V, $V_{cc} = 4.5$ V under 638 nm illumination with an optical power intensity of 113.579 mW/cm². The results are extracted from the corresponding time-revolved photo-response curves under different polarization angles (Supplementary Fig. 16). The relevant rectangular coordinate diagrams are shown in Fig. 3c. Owing to the amplification of the transistor, $I_{max}$ and $I_{min}$ are ~44.04 and 1.922 nA under 638 nm illumination. To better visualize the change of the polarized photocurrent change with the polarization angle, Formula 2 was used to fit the experimental data. According to the fitting result, the anisotropic photocurrent ratio is 87.3, which is one order of magnitude larger than that of a single $Bi_2Se_2S$ NW-based PS (1.32). Supplementary Fig. 17a–c shows similar experimental procedures and results of the integrated system under 808 nm illumination with an optical power intensity of 77.443 mW/cm². Compared with a single $Bi_2Se_2S$ NW-based PS, the anisotropic photocurrent ratio increased from 1.22 to 51.77. It can be seen from the fitting results that the polarization angle corresponding to the $I_{py}$ of the PSAS is similar to that of a single $Bi_2Se_2S$ NW-based PS, indicating that the transistor only plays a magnifying role in the PSAS, and does not affect the original polarization performance of the device. In addition to the above, the influence of polarized light on the photocurrent of the integrated system under 532 nm illumination and 37.714 mW/cm² power is described in Supplementary Fig. 18. The corresponding anisotropic photocurrent ratios are 375. More detailed experimental results for the PSAS under different illumination conditions are described in Supplementary Table 1. And the reason why $I_{px}$ is smaller at 532 nm illumination is probably due to the following two reasons: First, because PSAS is sensitive to changes for linearly polarized light, a small angle change can cause a large change in the photocurrent. Second, compared with 638 and 808 nm, the actual measured minimum photocurrent at 532 nm is close to the second-smallest photocurrent. Figure 3d shows a comparison of the obtained anisotropic photocurrent ratio with other start-of-the-art values reported in the literature for typical LD nanostructure-based devices. The specific parameters of these devices are shown in Supplementary Table 2[6,17–43]. It can be seen from the comparative results that although the anisotropic photocurrent ratio of a single $Bi_2Se_2S$ NW-based PS is not significantly higher than that of previously reported polarization-

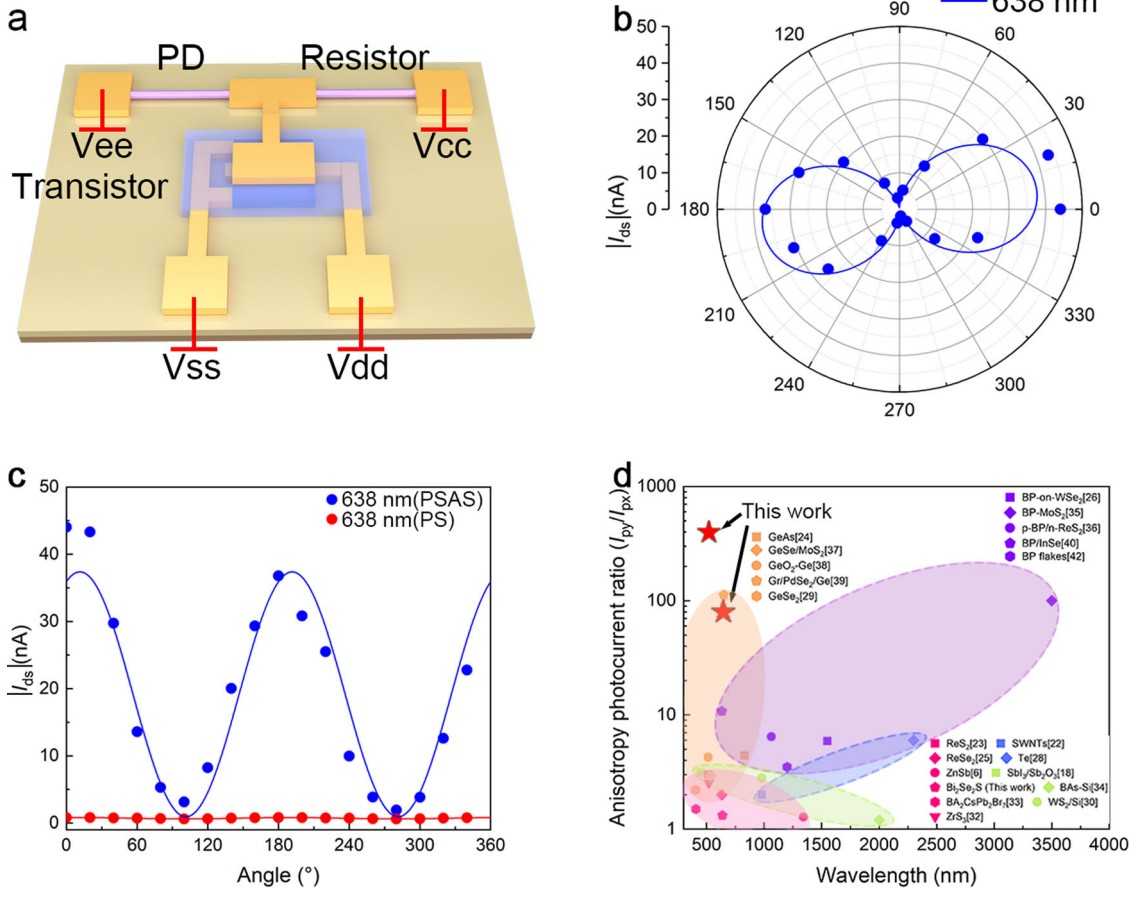

**Fig. 3 The polarized optoelectronic properties of PSAS. a** Structure schematic diagram of PSAS including $Bi_2Se_2S$ NWs based PS and reference resistor, the $C_8$-BTBT/PS-based OFET. **b** Polar plot of the relationship between photocurrent and polarization angle under 638 nm (optical power density of 113.579 $mW/cm^2$) at $V_{dd} = -0.5$ V and $V_{cc} = +4.5$ V. Circles and curves represent experimental data and the result of the formula fitting. **c** Rectangle coordinate of the relationship between photocurrent and polarization angle of PSAS and PS-based $Bi_2Se_2S$ NWs respectively under 638 nm. **d** Comparison of the anisotropic photocurrent ratio of our work and previous research results.

sensitive photodetectors based on LD materials, the performance is significantly improved after integration with the $C_8$-BTBT/PS OFET, which can be compared with the best research results. In addition, compared to most of the research work based on the intrinsic performance of LD and devices listed in Fig. 3d, our method is more universal. It not only has the ultra-high anisotropic photocurrent ratio (>20) of the nanoantenna method, but also has the wide-spectrum detection range of the photodetector based on LD materials. What's more, the manufacturing process is relatively simple. If the polarized photodetector has a higher anisotropic photocurrent ratio, better performance can be obtained through our PSAS. These results imply that our PSAS has excellent application potential in polarized light communication and polarized light imaging.

**Image processing and recognition**. The high anisotropic photocurrent ratio makes our PSAS more sensitive to changes in polarized light. When applied to polarized light imaging, it is easier to identify accurate information from complex background noise[44–47]. Thus, based on the two fitting curves of the photocurrent with the polarization angle shown in Fig. 3c, we constructed two images databases containing images with six letters (B, D, I, J, O, T), which represent the polarization images sensed by the $Bi_2Se_2S$ NW-based PS and the PSAS (Supplementary Fig. 19). Then, the two image databases were imported into the ANN for image recognition, and the results were compared. The process of building the database and ANN are

depicted in detail in Supplementary Note 5, and the structure of the ANN is shown in Fig. 4a. Figure 4b displays the recognition rates of the two databases, which confirms that compared with $Bi_2Se_2S$ NW-based PS, the perception and recognition accuracy of polarization images can be significantly improved using the PSAS. The recognition accuracy rate of 99% decreased from 573 training epochs to 27 training epochs. This 20-fold increase illustrates that clearer images can significantly increase the processing speed and reduce power consumption. To further reflect the advantages of the PSAS, we extracted parts of the images from the two databases, and input them into two trained neural networks for recognition. Then, we calculated the probability of each image yielding six possible outcomes. Figure 4c displays the results of the PSAS and $Bi_2Se_2S$ NW-based PS, respectively. Compared with the misjudgments that often occur when using $Bi_2Se_2S$ NW-based PS, the probability of the PSAS predicting the correct result is significantly improved. Therefore, the PSAS can more effectively find local polarization changes in the optical pattern, revealing more pattern details.

**Polarized light information detection and processing**. Owing to the excellent anisotropic photocurrent ratio, the PSAS structured by $C_8$-BTBT/PS OFET and $Bi_2Se_2S$ NW-based PS can easily distinguish polarized light in a variety of polarization angle ranges without error. As a proof-of-concept, we connected the system to the designed circuit to demonstrate the transmission of information through polarized light and used our PSAS for detection to realize

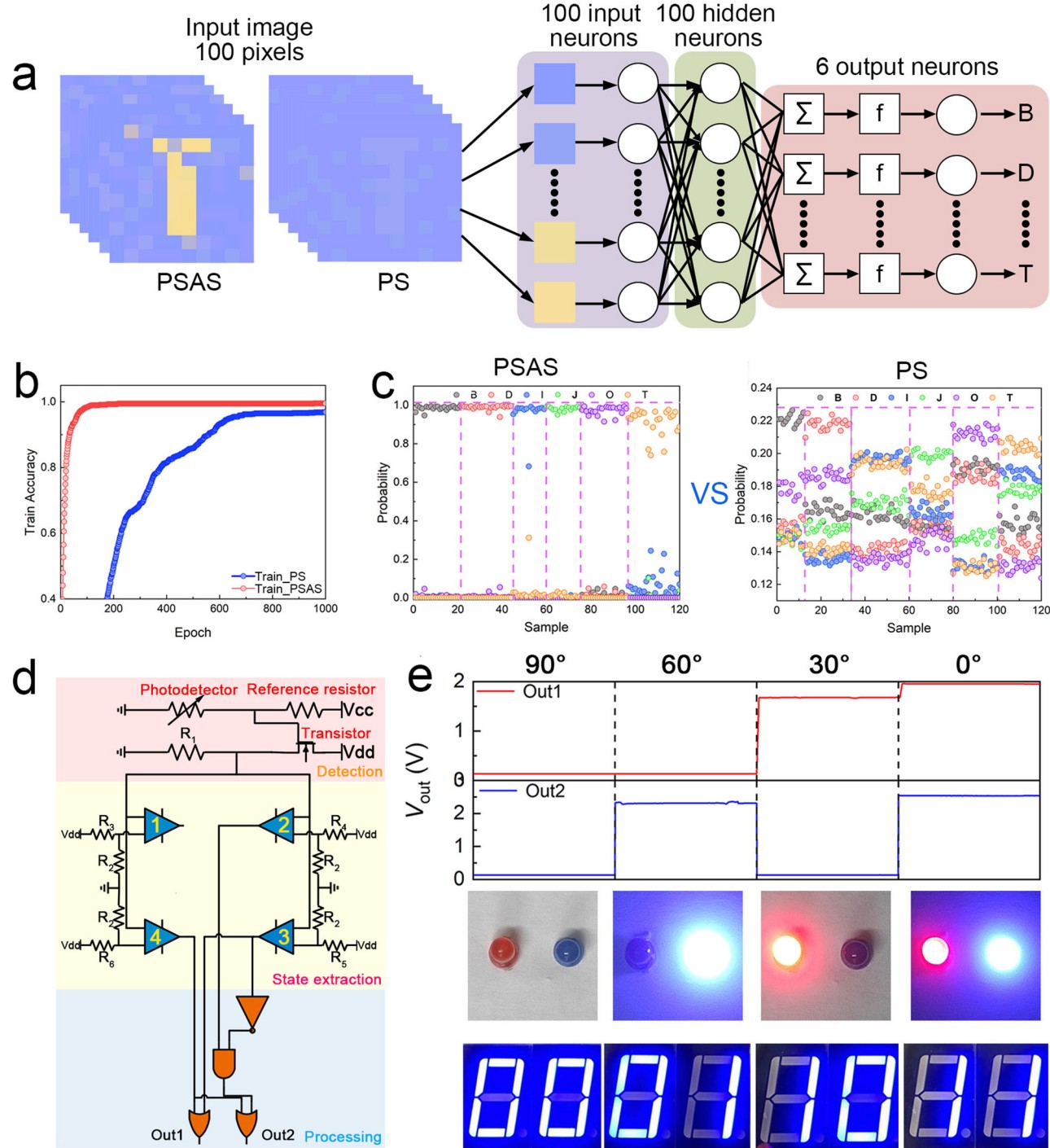

**Fig. 4 Simulations of polarization image recognition and polarized light information detection and processing based on PSAS. a** Illustration of polarization image recognition system based on the PSAS and $Bi_2Se_2S$ NW-based PS. The neural network is used in the processing and recognition part. **b** Comparisons of the image recognition accuracy with PSAS or $Bi_2Se_2S$ NW-based PS detecting image. **c** Probability of six possible results after inputting image example of PSAS and PS detected into a trained neural network. **d** The circuit diagram and system-level logical diagram of the entire system, including three parts: detection, state extraction, and processing. **e** The output voltage and coded information of the system after receiving different angels of polarized light.

information disguising. Figure 4d shows the circuit diagram of the polarized light information decoding system, which includes three parts: detection, state extraction, and processing. The PSAS is connected to the detection part to detect the polarized light, and a reference resistor is matched at the source electrode of the transistor to construct a voltage divider circuit. The aim is to convert the angle

change of the polarized light into the resistance change of the transistor and then into the output voltage change. Four voltage comparators are used to construct the state extraction part to determine five voltage ranges, and it is used to judge which range the voltage from the detection part belongs to. After the judgment is completed, the comparators output the voltages according to the voltage range,

and these are passed to the processing unit. The processing unit completes the decoding of the polarization angle of the polarized light through a certain logic operation and outputs the corresponding voltage through Out1 and Out2 to interpret the polarized light information. The output voltages of Out1 and Out2 are classified into two voltage ranges, corresponding to the two states of 0 and 1. Therefore, the combination of Out1 and Out2 can represent a two-digit binary number, and realize the decoding of four types of polarization information. To visualize the output results, two LED lights of different colors are connected to Out1 and Out2, respectively, as shown in Supplementary Fig. 20. The output results are obtained by the alternate flashing of the lights, and the polarization angle of the input polarized light is read out from them. It can be observed from Fig. 4e that as the polarization angle of the input polarized light changes from 90° to 0°, the output voltage of Out1 and Out2 also change, thereby driving the LED and digital tube to output different results, including four states: "00", "01", "10", "11". This illustrates that our PSAS can clearly and accurately identify the polarization angle changes of the four types of polarized light. This means that each beam of detected polarized light can be transmitted and interpreted as four types of messages. Since the probability of getting each message is the same, the corresponding transmission information volume is 4 bits, which means that the efficiency of information transfer is higher. The results indicate that the application of our PSAS to polarized light communication can not only maintain a higher information content but can also effectively utilize the characteristics of polarized light to realize information disguising and increase confidentiality.

## Discussion

In conclusion, we designed and constructed a PSAS that can amplify the anisotropic photocurrent ratio of polarization-sensitive photodetectors. The introduction of the $C_8$-BTBT/PS-based OFET into the PSAS can effectively extend the anisotropic photocurrent ratio of the $Bi_2Se_2S$ NW-based PS by two orders of magnitude. In addition, this simple and effective strategy leads to significant improvement in the anisotropic photocurrent ratio of other wavelengths and is suitable for almost all polarization-sensitive photodetectors. The high anisotropic photocurrent ratio makes it easier for the PSAS to distinguish the change in the polarization angle and connect with the back-end circuit. Leveraging this, we construct a simple polarized light communication system based on this PSAS, which achieves polarized light coding of information and digital output of polarized light information. These results indicate that by increasing the anisotropic photo-current ratio through integrated OFETs, the PSAS not only simplifies the polarized light detection system but also realizes the digital output of polarized light information, which shows significant potential for applications in the fields of polarized light communication and polarized light imaging.

## Methods

**Synthesis of $Bi_2Se_2S$ NW**. An alumina boat containing 100 mg of $Bi_2O_3$ powder was placed in the center of the quartz tube, which is in the heating zone of the tube furnace. Another alumina boat containing 1 g of mixture power was placed upstream of the quartz tube, which included 500 mg of Se and 500 mg of S. The temperature of this place held room temperature during growth. Subsequently, an alumina boat with a Si/SiO2 substrate was placed downstream of the quartz tube about 11–12 cm from the center of the tube furnace. The choice of carrier gas in the growth process of material is Ar gas. Before the material grows, set the maximum airflow rate for 30 min to drain the oxygen in the quartz tube. Then, the airflow was adjusted flow to 30 sccm (standard cubic centimeters per minute). During material growth, the temperature of the tube furnace was firstly rapidly increased from room temperature to 650 °C within 30 min, and then kept at this temperature for 10 min. When the temperature reached 600 °C, the alumina boat with mixture powder was pushed into the heating zone of the tube furnace. After the growth of the material, the $Bi_2Se_2S$ NWs deposited on the Si/SiO2 substrate can be observed through the microscope.

**$C_8$-BTBT/PS-based field-effect transistor fabrication**. Flexible OTFT devices were fabricated on PEN substrates. Ag electrodes were first thermally evaporated through shadow masks to define the gate electrode. To form the gate dielectric, poly (4-vinylphenol) and poly (melamine-co-formaldehyde) were dissolved in propylene glycol monomethyl ether acetate at a ratio of 20 and 10 wt% and spin-coated onto the substrates. This was followed by annealing at 150 °C for 1 h under ambient conditions. Next, Ag electrodes were thermally evaporated as source/drain electrodes with a channel width of 5000 µm and a channel length of 50 µm. Subsequently, the S/D electrodes were modified with pentafluorobenzenethiol. An organic semiconductor solution made by mixing $C_8$-BTBT and PS at a concentration of 10 mg mL$^{-1}$ in anisole (3:1 ratio by volume) was spin-coated to form the semiconducting layer, followed by annealing at 80 °C for 30 min. Finally, the devices were encapsulated with the supplied CYTOP without dilution and annealed at 80 °C for 30 min before the measurements.

**$Bi_2Se_2S$ NW-based PS and PSAS fabrication**. The photolithography process was utilized to fabricate the polarized photodetectors. First, the dry transfer technology was utilized to transfer $Bi_2Se_2S$ NWs to the PET substrate. Then, a Cr/Au patterned film (5/50 nm), serving as the electrodes at both ends of the polarized photo-detector and reference resistor, was deposited on the above substrate and $Bi_2Se_2S$ NWs. Finally, we connected one end of the polarized photodetector and the reference resistor, respectively, to the gate of the transistor based on C8-BTBT/PS through gold wires to complete the preparation of the PSAS.

**Device characterization and measurement**. The characteristic of the PSAS and the $Bi_2Se_2S$ NW-based PS were measured through a semiconductor characterization system (B1500, Keysight). The visible and IR illumination sources were power-adjustable light source capable of outputting homogeneous light. All characteristic tests of these devices were performed at room temperature. For carrying out the polarized light detection test, we built a set of optical path systems, which included a polarizer (Glan-Taylor prism), a half-wave plate, and a vertical sample stage for device placement. Before the laser was irradiated to the devices, it must be modulated by the polarizer and half-wave plate. We can change the polarization direction of polarized light by rotating the half-wave plate.

**Material characterization**. A TEM with an EDS analyzer (JEM-2010F) and a SEM (Zeiss Supra55 (VP)) were utilized to characterize the lattice structure, size, and morphologies of the $Bi_2Se_2S$ NW, respectively. The crystallinity of the Bi2Se2S NW was obtained by a powder XRD (Rigaku D/Max-2550, $\lambda = 1.5418$ Å) analyzed. A Microscopic confocal laser Raman spectrometer (InVia, Renishaw, excited by the 532 nm laser) was utilized to measure the angle-resolved Raman spectra of $Bi_2Se_2S$ NW.

## Data availability

The authors declare that the data that support the findings of this study are available within the paper and Supplementary Information files, and are available from the corresponding author upon request. The data are also available in figshare with the identifier (https://doi.org/10.6084/m9.figshare.16592471). Source data are provided with this paper.

## Code availability

The codes used for simulation and data plotting are available from the corresponding authors upon request and are also available in Zenodo (https://doi.org/10.5281/zenodo.5497748).

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

## Acknowledgements

This work was supported by the National Science Foundation of China (NSFC, Grant No. 62022079, 61625404, 61888102, 62125404, and 61874111), the National Key Research and Development Program of China (Grant No. 2017YFA0207500), and the Strategic Priority Research Program of the Chinese Academy of Sciences (Grant No. XDB43000000).

## Author contributions

W.R., Z.L., Z.W., and G.S. designed the research. W.R. and L.W. wrote the paper. W.R., Z.R., P.W., Y.Y., K.Z., L.L., Z.L., J.Y., and L.W. performed the experiments. W.R. and Z.L. analyzed the data. Z.L., Z.W., and G.S. supervised the project. All authors substantially contributed to the research and reviewed the manuscript.

## Competing interests

The authors declare no competing interests.
