## [Peer Review File · Nature Communications]

Integrated polarization-sensitive amplification system for digital information transmissionREVIEWER COMMENTS

Reviewer #1 (Remarks to the Author):

In this manuscript, the authors designed an interesting system by integrating polarized photodetector with organic transistor. It is remarkable and amazing that the anisotropy photocurrent ratio is enhanced by one to two orders of magnitude higher than the state-of-the-art low-dimensional nanostructures based polarized photodetectors. The dramatically enhanced anisotropy photocurrent ratio makes it possible to realize simple polarized light communication and coding, which expands the potential application range of polarized photodetectors. The idea in this work is attractive and the paper was well organized. I would like to recommend it for the publication in Nature Communications after addressing the following comments.

1. The fabrication process of the polarized photodetector should be clearly demonstrated.
2. The authors used organic transistors to amplify the polarization photodetectors. It is known that there are many factors that affect the performance of transistors, such as the selection of static operation point, the selection of voltage divider resistance, etc. The authors need to give some comments on these factors.
3. Figure 3d compared the anisotropic photocurrent ratio of different photodetectors. The authors are suggested to give some comments.
4. The authors should give some comments on the selection of the current organic transistor. What are the basic requirements for the performance of organic transistors?
5. XRD image in Supplementary Figure 1a should be clearly indexed.
6. SEM image in the supplementary Figure 12 lacks a scale bar.

Reviewer #2 (Remarks to the Author):

1. Why does the PSAS have a significantly low I_{px} at 532 nm illumination (reduced by order of magnitude, Supplementary Table 1) while I_{px} at 638 nm and 808 nm illumination remained similar to the PS result?
2. What is the purpose of choosing a C8-BTBT/PS OFET over other types of FET for the PSAS?

3. What are the specific advantages of using Bi₂Se₂S NW as the polarized photodetector? Is it due to the good anisotropic photocurrent ratio from the Bi₂Se₂S NW or the concept of PSAS? Please specify.
4. In principle, the PSAS has other disadvantages comparing with PS system. For example, responding time to the light might significantly deteriorate because the system has one more OFET.
5. In Fig. 2b, the authors claimed that the absorption coefficient of Bi₂Se₂S is significantly different along the x, y, and z directions from 200 nm to 1000 nm, but when the wavelength is 638nm, the coefficient of x and y directions are almost the same. It is not consistent with the later experiments data, which shows when apply 638 nm irradiation, linearly polarized light with different polarization angles exhibits different photoelectric.
6. There are some reports to directly fabricate photo-transistors with channels fabricated by semiconductors with anisotropic properties. What is the advantage of the reported device configuration compared with these works?
7. The illustration of ANN in Figure 4b is not correct. There should be at least one hidden layer added between the input neuron and output result.
8. It is interesting that the number of training epochs decreased by using the PSAS system. It is suggested that the authors should compare the data with/without PSAS system not only from the training result of ANN but also from the data itself.

Reviewer Comments to Author:

Reviewer 1

In this manuscript, the authors designed an interesting system by integrating polarized photodetector with organic transistor. It is remarkable and amazing that the anisotropy photocurrent ratio is enhanced by one to two orders of magnitude higher than the state-of-the-art low-dimensional nanostructures based polarized photodetectors. The dramatically enhanced anisotropy photocurrent ratio makes it possible to realize simple polarized light communication and coding, which expands the potential application range of polarized photodetectors. The idea in this work is attractive and the paper was well organized. I would like to recommend it for the publication in Nature Communications after addressing the following comments.

Response: Thank you for the Reviewer's constructive remark. Below, we respond individually to each specific point raised by the Reviewer and discuss how we have incorporated the Reviewer's suggestions into the revised manuscript.

Comments:

1. The fabrication process of the polarized photodetector should be clearly demonstrated.

Response: Thank you for Reviewer's good suggestion. We have supplementary the preparation process of the polarized photodetectors in the revised manuscript.

Revised (Page 25 line 7-15):

Bi₂Se₂S NW based PS and PSAS fabrication. Photolithography process was utilized to fabricate the polarized photodetectors. First, the dry transfer technology was utilized to transfer Bi₂Se₂S NWs to PET substrate. Then, a Cr/Au patterned film (5/50 nm), serving as the electrodes at both ends of the polarized photodetector and reference resistor, was deposited on the above substrate and Bi₂Se₂S NWs. Finally, we connected one end of the polarized photodetector and the reference resistor, respectively, to the gate of the transistor based on C₈-BTBT/PS through gold wires to complete the preparation of the PSAS.

2. The authors used organic transistors to amplify the polarization photodetectors. It is known that there are many factors that affect the performance of transistors, such as the selection of static operation point, the selection of voltage divider resistance, etc. The authors need to give some comments on these factors.

Response: We thank the Reviewer for the constructive remark. The transistor that can be used to amplify the detection signal must have a small sub-threshold swing and a small threshold voltage, like the OFET based on C₈-BTBT/PS we used, where the sub-threshold swing is 160 mV/decade and the threshold voltage is about 1.3 V. Then, the transistor amplification model is divided into large signal model and small signal model. Since we want to use OFET to realize the polarization photodetector to achieve anisotropic photocurrent ratio amplification, the large signal model is a very suitable choice. This is because the gain of the large signal model changes with the change of the gate voltage. In order to make the OFET work in a large signal model, the gate voltage must have a large swing with the input signal. Therefore, we choose a device with a resistance similar to that of the polarization photodetectors as the reference resistor (Bi₂Se₂S NW as well). To make

full use of the performance of OFET, we choose the static operation point at the threshold voltage of 1.3 V of the transistor. The specific model setting of PSAS are listed in Table R1. The relevant comments have been given in the revised manuscript.

Table R1 The specific model setting of PSAS.

Signal Model	Static Operation Point	Reference Resistor
Large signal model	$V_{gs} = 1.3 \text{ V}$	$\text{Bi}_2\text{Se}_2\text{S NW}$

Revised (Page 16-17):

The system mainly uses the large signal model of transistors. The transistor with low threshold voltage and low sub-threshold swing is very important for our system. The detailed electric properties of the OFET are described in Supplementary Note 4. C₈-BTBT/PS OFET with 1.3 V threshold voltage and 160 mV/decade sub-threshold swing can meet the requirements of PSAS. While realizing the functions, it makes the whole system work at a lower voltage. The response time of less than 25 ms based on the frequency response characteristics makes the C₈-BTBT/PS OFET match well with the Bi₂Se₂S NW-based PS.

3. Figure 3d compared the anisotropic photocurrent ratio of different photodetectors. The authors are suggested to give some comments.

Response: We thank the Reviewer for the constructive remark. For most of the research work listed in Figure 3d, the anisotropic photocurrent ratio of them are the intrinsic performance of the material or device, and the device structure mostly uses heterojunction or intrinsic materials. We detail the advantages and disadvantages of several common methods below. 1) Nanoantenna: Its biggest advantage is the use of nano-metal antenna structure design to achieve ultra-high anisotropic photocurrent ratio (> 20). Compared with its advantages, the disadvantages of this method are also quite obvious. The fixed structural design results in a narrow applicable wavelength range. Moreover, the shorter the wavelength you want to apply, the smaller the sized of the nanoantenna, which makes the manufacturing process difficult and complicated. 2) The intrinsic performance of the material or device: By using the anisotropy of the low-dimensional material lattice structure to detect polarized light, the detection can be realized in a wide spectral range, and the device is relatively simple to manufacture. However, the low anisotropic photocurrent ratio (<10), limits its practical application. Although a large anisotropic photocurrent ratio (> 100) was once reported in black phosphorus at a wavelength of 3.5 μm, the complex device structure design and precise energy band matching greatly increase the difficulty of manufacturing the entire device.

On the contrary, it is a universal method for us to realize the improvement of the anisotropic photocurrent ratio by constructing an integrated system. Using our method, the original anisotropic photocurrent ratio of Bi₂Se₂S NW polarized photodetector can be increased from 1.24 to 375. If the polarized photodetector has a higher anisotropic photocurrent ratio, better performance can be obtained through our integrated system. Such comment has been provided in the revised manuscript.

Revised (Page 18, line 14-21):

In addition, compared to most of the research work based on the intrinsic performance of LD and devices listed in Figure 3d, our method is much more universal. It not only has the ultra-high anisotropic photocurrent ratio (> 20) of the nanoantenna method, but also

has the wide-spectrum detection rang of the photodetector based on LD materials. What's more, the manufacturing process is relatively simple. If the polarized photodetector has a higher anisotropic photocurrent ratio, better performance can be obtained through our PSAS.

4. The authors should give some comments on the selection of the current organic transistor. What are the basic requirements for the performance of organic transistors?

Response: We thank the Reviewer for the constructive remark. Sub-threshold swing, threshold voltage, on/off current ratio and speed are the most important indicators for judging OFET performance. OFET with excellent performance must have low sub-threshold swing, low threshold voltage, high on/off current ratio and high speed. If we need to use the large signal model of transistors to achieve amplification similar to the anisotropic photocurrent ratio in this study, compared to the speed requirements, low sub-threshold swing, low threshold voltage and high on/off current ratio are more important. The better these parameters are, the better function of the integrated system will be. In addition, the solution preparation process of OFET makes the production of the device relatively simple and easy to produce on a large area. The flexible substrate makes our PSAS more flexible in the application process and can adapt to complex environments. The main reason why we choose C8-BTBT/PS OFET is that its various parameters meet our requirements, which is shown in Figure R1. If there are other FETs that also meet the above requirements, they can be used to replace the C8-BTBT/PS OFET used in PSAS.

Figure R1 The various parameters of OFET, including V_{th} , SS, On/Off ratio and Zero potential current (I_{ds} , $V_g = 0V$).

Revised (Page 16-17):

The system mainly uses the large signal model of transistors. The transistor with low threshold voltage and low sub-threshold swing are very important for our system. The detailed electric properties of the OFET are described in Supplementary Note 4. C8-BTBT/PS OFET with 1.3 V threshold voltage and 160 mV/decade sub-threshold swing can meet the requirements of PSAS. While realizing the functions, it makes the whole system work at a lower voltage. The response time of less than 25 ms based on the frequency response characteristics makes the C8-BTBT/PS OFET match well with the Bi_2Se_2S NW-based PS. In addition, the solution preparation process of OFET makes the production of the device relatively simple and easy to produce on a large area. The flexible

substrate makes our PSAS more flexible in the application process and can adapt to complex environments.

5. XRD image in Supplementary Figure 1a should be clearly indexed.

Response: We thank the Reviewer for the constructive remark. The XRD image in Supplementary Figure 1a has been corrected.

Revised (Supplementary Figure 1):

Supplementary Figure 1 a XRD image and **b** EDS image of $\text{Bi}_2\text{Se}_2\text{S}$ NWs.

6. SEM image in the supplementary Figure 12 lacks a scale bar.

Response: We thank the Reviewer for the constructive remark. The SEM image in Supplementary Figure 12 has been corrected.

Revised (Supplementary Figure 13):

Supplementary Figure 13 The physical diagram of the PSAS.

Reviewer 2

Comments:

1. Why does the PSAS have a significantly low I_{px} at 532 nm illumination (reduced by order of magnitude, Supplementary Table 1) while I_{px} at 638 nm and 808 nm illumination remained similar to the PS result?

Response: We thank the Reviewer for the constructive remark. I_{px} is a fitting parameter, which is obtained by fitting the polarized photocurrent with the polarization angle of linearly polarized light through the formula (Malus Law): $I_{ph}(\delta) = I_{px}\sin^2(\delta + \Phi) + I_{py}\cos^2(\delta + \Phi)$. It can be seen from the fitting results that I_{px} and I_{py} correspond to the theoretical minimum and maximum polarized photocurrent, respectively. The fitting formula is used to fit the test results because we measure a set of polarized photocurrent for every 20° change in the polarization angle of linearly polarized light during the measurement process, which can easily cause us to miss the minimum polarized photocurrent and the maximum polarized photocurrent, especially for PSAS with sensitive response for the changes of polarization angle which can better reflect the change trend of polarized photocurrent. Many researches use this formula to fit the test results¹⁻⁴. We use the above formula to fit the anisotropic photocurrent of the PSAS at 532 nm illumination, and the obtained I_{px} is listed in the Table S1 with the high fitting result of 96.88%. The reason why I_{px} is smaller at 532 nm illumination is probably due to the following two reasons: 1) First, because PSAS is sensitive to changes for linearly polarized light, a small angle change can cause a large change in the photocurrent. 2) Second, compared with 638 nm and 808 nm, the actual measured minimum photocurrent at 532 nm is close to the second-smallest photocurrent. Therefore, the angle corresponding to I_{px} should be close to the middle of the angles corresponding to the two photocurrents. Since the two angles differ by 20°, the difference between I_{px} and the angle corresponding to the minimum photocurrent obtained by the test is close to 10°. Therefore, I_{px} is much smaller than the minimum photocurrent at 532 nm. On the contrary, the actual test minimum photocurrent and the second-smallest photocurrent are quite different at 638 nm and 808 nm, so the fitted I_{px} should be close to the minimum photocurrent. Therefore, I_{px} at 532 nm is much smaller than that of at 638 nm and 808 nm, and higher than the off-state current of C₈-BTBT/PS OFET which is corresponding well with the results.

1. Chen Y, *et al.* Ferroelectric-tuned van der Waals heterojunction with band alignment evolution. *Nat. Commun.* 12, 4030 (2021).
2. Tong L, *et al.* Stable mid-infrared polarization imaging based on quasi-2D tellurium at room temperature. *Nat. Commun.* 11, 2308 (2020).
3. Wang J, *et al.* Ultrasensitive polarized-light photodetectors based on 2D hybrid perovskite ferroelectric crystals with a low detection limit. *Sci. Bull.* 66, 158 (2021).
4. Yang Y, *et al.* Polarization-Sensitive Ultraviolet Photodetection of Anisotropic 2D GeS₂. *Adv. Funt. Mater.* 29, 1900411 (2019).

2. What is the purpose of choosing a C₈-BTBT/PS OFET over other types of FET for the PSAS?

Response: We thank the Reviewer for the constructive remark. We choose C₈-BTBT/PS OFET for the PSAS based on the following two reasons. 1) First, organic transistors such as C₈-BTBT/PS OFET are very suitable for flexible devices. The solution method is used

to prepare transistors, the whole process is relatively simple and easy to prepare large areas. What's more, the device uniformity is better. 2) Second, using the large signal model for the transistor to achieve amplification similar to the anisotropic photocurrent ratio in this research requires high performance of the transistor, such as low sub-threshold swing, low threshold voltage and high on/off current ratio. The better these parameters are, the more obvious the amplification effect of the integrated system will be. After investigation the existing research work of OFET, we find that the parameters of the C₈-BTBT/PS OFET meet our needs. It can be found from the test results that the sub-threshold swing of C₈-BTBT/PS OFET is 160 mV/decade, the threshold voltage is 1.3 V and the on/off current ratio is 10⁵. When the gate voltage reaches 0 V, the output current is already close to saturation and meets our needs, so we choose C₈-BTBT/PS OFET. The specific parameters are listed in Table R2. The reason for choosing C₈-BTBT/PS OFET has been given in the revised manuscript. If there are other transistors, whether they are parameters or compound requirements in preparation, they can also be applied to PSAS. The reasons of choosing C₈-BTBT/PS OFET and the influence of C₈-BTBT/PS OFET's response time for PSAS have been provided according to the reviewer's suggestion.

Table R2 The specific parameters of OTFT

Material	Threshold Voltage (V _{th})	Sub-threshold Swing (SS)	On/Off Ratio	Zero potential current (I _{ds} , V _g = 0V)
C ₈ -BTBT/PS	1.3 V	160 mV/decade	>10 ⁵	-300 nA (V _{ds} = -0.5 V)

Revised (Page 16-17):

The system mainly uses the large signal model of transistors. The transistor with low threshold voltage and low sub-threshold swing are very important for our system. The detailed electric properties of the OFET are described in Supplementary Note 4. C₈-BTBT/PS OFET with 1.3 V threshold voltage and 160 mV/decade sub-threshold swing can meet the requirements of PSAS. While realizing the functions, it makes the whole system work at a lower voltage. The response time of less than 25 ms based on the frequency response characteristics makes the C₈-BTBT/PS OFET match well with the Bi₂Se₂S NW-based PS. In addition, the solution preparation process of OFET makes the production of the device relatively simple and easy to produce on a large area. The flexible substrate makes our PSAS more flexible in the application process and can adapt to complex environments.

3. What are the specific advantages of using Bi₂Se₂S NW as the polarized photodetector? Is it due to the good anisotropic photocurrent ratio from the Bi₂Se₂S NW or the concept of PSAS? Please specify.

Response: We thank the Reviewer for the constructive remark. As a topological insulator, Bi₂Se₃ is identified with the surface state consisting of a single Dirac cone. It can be a state of matter that may serve as a platform for both fundamental physics phenomena and technological applications, such as spintronics and quantum information processing⁵. These interesting properties attract us to study Bi-based S compounds such as Bi₂Se₂S. In Supplementary Note 1, we describe the material structure of Bi₂Se₂S NWs in detail, including lattice structure, etc. Bi₂Se₂S NWs belong to orthorhombic structure, which is a low-symmetry lattice structure. This low symmetry crystal structure makes Bi₂Se₂S often grow into a NW morphology, so there is bound to be anisotropy in photoelectric properties.

The results of angle-resolved polarized Raman spectroscopy (ARPRS) further confirm the anisotropy of Bi₂Se₂S NWs in crystal structure. Due to the low symmetry of the lattice structure, Bi₂Se₂S NWs are suitable for preparing polarized photodetectors. However, in the actual test process, we found that the anisotropic photocurrent ratio of the Bi₂Se₂S NW-based polarized photodetector was not enough to support our subsequent experiments on polarized light communication. So, we designed the PSAS on this basis, it is used to increase the anisotropic photocurrent ratio, so as to realize the subsequent experiments of polarized light communication. The advantages of using Bi₂Se₂S NW have been specified in the revised manuscript.

5. Chen YL, et al. Massive dirac fermion on the surface of a magnetically doped topological insulator. Science 329, 659 (2010).

Revised (Page 9, line 4-12):

Bi₂Se₂S NWs were synthesized by chemical vapor deposition and the crystal quality and elemental composition were verified through various material characterizations (Supplementary Note 1). According to the characterization results, we can find that the Bi₂Se₂S NWs belong to the orthorhombic structure, which is a low-symmetry lattice structure. This low symmetry crystal structure makes Bi₂Se₂S easily grow into a NW morphology, therefore there must be a primary growth direction and a secondary growth direction in the growth plane. This will cause the absorption coefficient of the incident light to be different in the two direction, so there is bound to anisotropy in photoelectric properties.

4. In principle, the PSAS has other disadvantages comparing with PS system. For example, responding time to the light might significantly deteriorate because the system has one more OFET.

Response: Thank you for Reviewer's good suggestion. According to existing research reports, for polarization-sensitive photodetectors to be used in practice and extract useful polarization information, the anisotropic photocurrent ratio must be higher than 20^{6,7}. This index cannot be achieved by many polarization-sensitive photodetectors based on PS. Compared with the PS system, PSAS has a huge advantage, that is, it can increase the anisotropic photocurrent ratio by several orders of magnitude, which can reach this target easily. The PSAS have some disadvantages, like complex design, complex preparation process, et al. However, owing to the high anisotropic photocurrent ratio, PSAS does not require a polarizer like in PS, and reduces the need for the filtering and amplifying circuits. In turn, the system structure can be optimized substantially, and the complexity of the system design can be reduced, and effectively simplify the structure and processing of the entire polarized optical communication system.

For responding time of PSAS, according to the frequency response characteristics of the Supplementary Figure 14, we can determine that the response time of OFET must be less than 25 ms. In our work, the rise time and fall time of the PS based on Bi₂Se₂S NWs are >400 ms and <400 ms, respectively, which are shown in Supplementary Figure 12. Therefore, the device that ultimately determines the operating speed of the entire system is not the OFET, but the Bi₂Se₂S NW based polarization photodetector. So, the response time of the PS system and the PSAS is basically the same.

6. Wei J, *et al.* Mid-infrared semimetal polarization detectors with configurable polarity transition. *Nat. Photon.* 3052 (2021).
7. Gruev V, *et al.* polarization imaging sensor with aluminum nanowire optical filters. *Opt. Exp.* 18, 19087-19094 (2010).

Revised (Page 15, line 4-5 and Supplementary Figure 12):

Supplementary Figure 12 displays the responding time of Bi₂Se₂S NW based PS ($\tau_{\text{rise}} > 400$ ms, $\tau_{\text{decay}} < 400$ ms).

Supplementary Figure 12 The responding time of Bi₂Se₂S NW based PS.

5. In Fig. 2b, the authors claimed that the absorption coefficient of Bi₂Se₂S is significantly different along the x, y, and z directions from 200 nm to 1000 nm, but when the wavelength is 638nm, the coefficient of x and y directions are almost the same. It is not consistent with the later experiments data, which shows when apply 638 nm irradiation, linearly polarized light with different polarization angles exhibits different photoelectric.

Response: We thank the Reviewer for the constructive remark. This is mainly because the unit cell base vector we used in the theoretical calculation is different from the based vector in XRD and TEM. We use XRD to determine the TEM crystal plane using the base vector is the Bi₂Se₂S (JCPDS Card No. 19-0174, $a = 11.5 \text{ \AA}$, $b = 11.7 \text{ \AA}$, $c = 4.073 \text{ \AA}$, $\alpha = 90^\circ$, $\beta = 90^\circ$, $\gamma = 90^\circ$). Based on the crystal planes calibrated in TEM, we judge that Bi₂Se₂S NWs grow along the a-axis and stack along the b-axis during the growth process. Therefore, when linearly polarized light is irradiated on the NW, the anisotropic photocurrent is mainly due to the different in the absorption of polarized light by the Bi₂Se₂S NW in the a-axis and b-axis direction. In theoretical calculations, the lattice model we used is the structure in the Materials Project lattice library ($a = 4.134 \text{ \AA}$, $b = 11.47 \text{ \AA}$, $c = 11.9 \text{ \AA}$, $\alpha =$

90°, $\beta = 90^\circ$, $\gamma = 90^\circ$). There is an angular deviation between the two base vectors. Therefore, the a-axis, b-axis and c-axis of the model in theoretical calculations correspond to the c-axis, a-axis and b-axis in the XRD card, respectively. The absorption of the a-axis and b-axis that play a major role in the anisotropic photocurrent analyzed above corresponds to the b-axis and c-axis in the theoretical calculation model. From the theoretically calculated absorption spectrum in Figure 2b, it can be seen that both the b-axis (y-axis) and the c-axis (z-axis) have obvious absorption anisotropy at the wavelength (532 nm, 638 nm, 808 nm) we tested. In order to show the results more clearly, we changed the base vector label of the theoretical calculation to the base vector of the XRD card and revised the label in Figure 2b, Supplementary Figure 2a and Supplementary Figure 4b, and calculated the ratio of the absorption coefficient on the a-axis and b-axis (α_a/α_b), which result is shown in Supplementary Figure 4c. In order to further prove that Bi₂Se₂S nanowires have strong anisotropy for light absorption on the a-axis and b-axis, we calculated the probability (p_a) of the transition from $|\varphi_{CBM}\rangle$ to $|\varphi_{VBM}\rangle$ along the k-points path, which calculation formula is $p_a = |\langle\varphi_{CBM}|\nabla_a|\varphi_{VBM}\rangle|^2 / (|\langle\varphi_{CBM}|\nabla_a|\varphi_{VBM}\rangle|^2 + |\langle\varphi_{CBM}|\nabla_b|\varphi_{VBM}\rangle|^2)$, where φ represents the wave function, ∇ represents the momentum operator and $|\langle\varphi_{CBM}|\nabla|\varphi_{VBM}\rangle|^2$ is the optical transition amplitude. According to the calculation results in Supplementary Figure 4d, we can clearly observe that only the absorption of a-polarized light occurs across the gap between Vb Γ and Cb Γ at the Γ point ($p_a = 1.0$), while the absorption of b-polarized light is forbidden. The discussion has been highlighted in the revised manuscript.

Revised (Page 15, line 4-5, Figure 2b, Supplementary Figure 4 and Supplementary Note 1):

The atom-resolved and total density of states (DOS) of the bulk Bi₂Se₂S are also shown in Supplementary Figure 4a, which indicates that S and Se atoms mainly contribute to the top of the valence band of Bi₂Se₂S, while Bi atoms mainly contribute to the bottom of the conduction band. To investigate the anisotropic absorption of bulk Bi₂Se₂S, the partial charge densities of the valence band maximum (VBM) and conduction band minimum (CBM) in the ab- and bc-planes were calculated by VASP, and are shown in Supplementary Figure 4b. It was found that the charges were mainly localized on the S and Se atoms at the VBM, while they were localized on the Bi atom at the CBM. This is in good agreement with the DOS results. Theoretically, the electric dipole transition probability of photons absorbed per unit time is proportional to the electron-radiation interaction Hamiltonian matrix element $|\langle\langle c|H_{eR}|v\rangle\rangle|$, where $|c\rangle$ and $|v\rangle$ are the wave functions of electrons in the conduction band and valence band, respectively. Therefore, different charge distributions along the a-, b-, and c-axes result in the anisotropic optical properties of Bi₂Se₂S. The optical absorption spectrum of Bi₂Se₂S was obtained using first-principle calculations. Based on the crystal planes calibrated in TEM, we can see that Bi₂Se₂S NWs grow along the a-axis and stack along the b-axis during the growth process. Therefore, when linearly polarized light is irradiated on the NW, the anisotropic photocurrent is mainly due to the different in the absorption of polarized light by the Bi₂Se₂S NW in the a-axis and b-axis direction. As shown in Figure 2b, the absorption coefficient of Bi₂Se₂S is significantly different along the a and b directions from 300 nm to 1000 nm. Supplementary Figure 4c displays the ratio of absorption coefficient of a-axis and b-axis (α_a/α_b), which illustrates a-axis and b-axis have obvious optical anisotropy. In order to further prove that Bi₂Se₂S NWs have strong anisotropy for light absorption on the

a-axis and b-axis, we calculated the probability (p_a) of the transition from $|\varphi_{CBM}\rangle$ to $|\varphi_{VBM}\rangle$ along the k-points path, which calculation formula is:

$$p_a = \frac{|\langle \varphi_{CBM} | \nabla_a | \varphi_{VBM} \rangle|^2}{(|\langle \varphi_{CBM} | \nabla_a | \varphi_{VBM} \rangle|^2 + |\langle \varphi_{CBM} | \nabla_b | \varphi_{VBM} \rangle|^2)}$$

where φ represents the wave function, ∇ represents the momentum operator and $|\langle \varphi_{CBM} | \nabla | \varphi_{VBM} \rangle|^2$ is the optical transition amplitude. According to the calculation results in Supplementary Figure 4d, we can clearly observe that only the absorption of a-polarized light occurs across the gap between $V_{b\Gamma}$ and $C_{b\Gamma}$ at the Γ point ($p_a = 1.0$), while the absorption of b-polarized light is forbidden.

Supplementary Note 1:

The crystals structure of the $\text{Bi}_2\text{Se}_2\text{S}$ is shown in Supplementary Figure 2a. It is evident that $(\text{Bi}_8\text{Se}_8\text{S}_4)_n$ is stacked along the a-axis through strong covalent Bi-S and Bi-Se bonds, and is held together by van der Waals forces in the b-axis. Therefore, at the edge of $\text{Bi}_2\text{Se}_2\text{S}$, the bonding force between atoms along the a-axis is stronger than that along the b-axis, which leading that the $\text{Bi}_2\text{Se}_2\text{S}$ tends to grow along the a-axis.

The secondary growth axis is the b-axis. When linearly polarized light is irradiated on the surface of the material, the structure anisotropy of the a-axis and b-axis of the $\text{Bi}_2\text{Se}_2\text{S}$ NW will be main factor affecting the anisotropic photocurrent of the material•••

Fig. 2 Material characterization and polarized optoelectronic response of $\text{Bi}_2\text{Se}_2\text{S}$ NW based PS. **a** The band structure of the bulk $\text{Bi}_2\text{Se}_2\text{S}$ calculated by the HSE06 functional. **b** Calculated optical absorption coefficients $\alpha(\omega)$ of $\text{Bi}_2\text{Se}_2\text{S}$ along the x, y and z axes. **c** The angle-resolved Raman scattering spectra under parallel configurations of 532 nm laser. **d** Polar plot of Raman peaks at 166 cm^{-1} belonging to the A_g mode under parallel configurations of 532 nm laser. **e** Current-voltage (I-V) curve under 638 nm (optical power density of 111.368 mW/cm^2) with different polarization angle. **f** Polar diagram of relationship between photocurrent and polarization angle under 638 nm at 1.0 V drain bias, the anisotropic photocurrent ratio is 1.32.

Supplementary Figure 4 **a** The atom-resolved and total density of states (DOS) of the bulk $\text{Bi}_2\text{Se}_2\text{S}$ **b** Calculated partial charge density of bulk $\text{Bi}_2\text{Se}_2\text{S}$ at the state of VBM (left) and CBM (right). The purple, yellow and green balls represent Bi, S and Se atoms, respectively. **c** The ratio of absorption coefficient of a-axis and b-axis (α_a/α_b). **d** The p_a of optical transition from $|\varphi_{\text{CBM}}\rangle$ to $|\varphi_{\text{VBM}}\rangle$ along the k-point path.

6. There are some reports to directly fabricate photo-transistors with channels fabricated by semiconductors with anisotropic properties. What is the advantage of the reported device configuration compared with these works?

Response: We thank the Reviewer for the constructive remark. In Figure 3d, we have listed a lot of existing research work, most of which are made of semiconductor with anisotropic characteristics. Among them, there are many excellent research work. However, these research works are not universal and rely heavily on the excellent properties of the materials themselves. It is indeed an excellent idea to use photo-transistor and use gate voltage (V_g) modulation to increase the anisotropic photocurrent and the anisotropic photocurrent ratio. However, this method may have a problem. The anisotropic photocurrent has been significantly improved, but the anisotropic photocurrent ratio is basically unchanged. Because changing the gate voltage of the photo-transistor will change the photocurrent as a whole, which will cause the anisotropic photocurrent to increase or decrease at the same time. Except for the significant changes in the anisotropic photocurrent, the anisotropic photocurrent ratio did not increase significantly. Therefore, it is a better way to use the adjustable gate voltage of the phototransistor to realize the improvement of the anisotropic photocurrent ratio, but the effect of only relying on the gate voltage modulation may not be obvious. May need to be supplemented by some material design, such as energy band structure design. On the contrary, it is a general method for us to realize the improvement of the anisotropic photocurrent ratio by constructing an integrated system. We can use this method to increase the anisotropic photocurrent ratio of the $\text{Bi}_2\text{Se}_2\text{S}$ NW polarized photodetector by several orders of

magnitude. If we replace the polarized photodetector used in the integrated system with a polarized photocurrent with better performance that has been reported, our integrated system will achieve even more excellent performance.

7. The illustration of ANN in Figure 4b is not correct. There should be at least one hidden layer added between the input neuron and output result.

Response: We thank the Reviewer for the constructive remark. When designing the ANN, we used a hidden layer containing 100 neurons. We have corrected the Figure 4a in the revised manuscript.

Revised (Figure 4):

Fig. 4 Simulations of polarization image recognition and polarized light information detection and processing based on PSAS. a Illustration of polarization image recognition system based on the PSAS and Bi₂Se₂S NW based PS. The neural network is used in processing and recognition part. **b** Comparisons

of the image recognition accuracy with PSAS or Bi₂Se₂S NW based PS detecting image. **c** Probability of six possible results after inputting image example of PSAS detected into a trained neural network. **d** Probability of six possible results after inputting image example of Bi₂Se₂S NW based PS detected into a trained neural network. **e** The circuit diagram and system-level logical diagram of the entire system, including three parts: detection, state extraction and processing. **f** The output voltage and coded information of the system after receiving different angels of polarized light.

8. It is interesting that the number of training epochs decreased by using the PSAS. It is suggested that the authors should compare the data with/without PSAS not only from the training result of ANN but also from the data itself.

Response: We agree with the Reiewer’s point of view. The accuracy and efficiency of image recognition have a great relationship with the clarity and resolution of the input images. ANN is earsier to extract features from distinct and high-resolution input images and make accurate judgements. In order to compare the different between the dataset constructed based on the PSAS and the dataset constructed by PS, we extracted partially constructed pictures from the two datasets, which are shown in Supplementary Figure 19. Supplementary Figure 19a shows six types of images in two datasets. Supplementary Figure 19b and c show images constructed according to the method of Supplementary Note 5 on the basis of six basic images, respectively. From Supplementary Figure 19b and 19c, it can be seen that the image constructed by the PSAS, in which the background noise has little effect on the letter pattern. On the contrary, under the influence of background noise, the letters in the image constructed from PS shown in Supplementary Figure 19b will be blurred a lot, which will cause the efficiency and accuracy of image recognition to decrease. Especially when we set the maximum and minimum values in the scale of the heat map to a different of 20 times, the letters are basically submerged in noise. A gap if 20 times is a basic requirement for polarization imaging and polarization communication. The detailed discussion is provided in Supplementary Note 5.

Revised (Supplementary Figure 19 and Supplementary Note 5):

Supplementary Figure 19 a Idea patterns of six letters (B, D, I, J, O, T). **b** Incomplete pattern with background noise based on PS system. **c** Incomplete pattern with background noise based on PSAS.

Finally, we get two image datasets, which belong to PSAS and Bi₂Se₂S based PS (Supplementary Figure 19). Supplementary Figure 19a shows six types of images in two datasets. Supplementary Figure 19b and c show images constructed according to the above method on the basis of six basic images, respectively. From Supplementary Figure 19b and 19c, it can be seen that the image constructed by the PSAS, in which the background noise has little effect on the letter pattern. On the contrary, under the influence of background noise, the letters in the image constructed from PS shown in Supplementary Figure 19b will be blurred a lot, which will cause the efficiency and accuracy of image recognition to decrease.

REVIEWERS' COMMENTS

Reviewer #1 (Remarks to the Author):

The authors have responded all the comments from this review. The manuscript is now suitable for publication in my opinion.

Reviewer #2 (Remarks to the Author):

The authors have properly addressed my comments. I recommend acceptance of this work for publication.

Reviewer Comments to Author:

Reviewer 1

The authors have responded all the comments from this review. The manuscript is now suitable for publication in my opinion.

Response: Thank you for the Reviewer's constructive remark.

Reviewer 2

The authors have properly addressed my comments. I recommend acceptance of this work for publication.

Response: Thank you for the Reviewer's suggestions and recognition.